# Phase transformation in lead titanate based relaxor ferroelectrics with ultra-high strain

Hangfeng Zhang[1], Zilong Li [1], Yichen Wang [1], A. Dominic Fortes [2],
Theo Graves Saunders[3], Yang Hao [3] ✉, Isaac Abrahams [4] ✉,
Haixue Yan [1] ✉ & Lei Su [1] ✉

The reverse piezoelectric effect allows for the conversion of an electrical input signal into mechanical displacement and forms the basis for the operation of positioners and actuators. Addressing the practical need for cost-effective sensitive materials, we introduce erbium-doped lead magnesium niobium titanate ceramics which exhibit exceptionally high strain (3.19% bipolar and 0.8% unipolar) under a very low applied field of 2 kV mm$^{-1}$, resulting in record-breaking piezoelectric coefficients ($d_{33}^*$ values of 15,950 and 4014 pm V$^{-1}$, respectively). These exceptional properties stem from a combination of factors including the sensitivity of polar nanoregions to the applied field in this relaxor ferroelectric system, the thickness of the sample, and the energetic availability of polymorphs with different polar structures where a change in polarisation direction occurs at the field induced phase transition. Surpassing the performance of single crystal materials, our findings establish a benchmark in piezoelectric performance with implications for many diverse applications.

The piezoelectric phenomenon exhibited by ferroelectric materials, forms the basis of devices such as actuators, ultrasound sensors, and energy harvesters[1–3], which are extensively used in diverse applications ranging from medical therapeutics and diagnostics to electric vehicles[4–6]. Piezoelectricity can be characterised by field-induced strain, denoted as the ratio $\Delta l/l$, where $l$ represents the thickness of the material and $\Delta l$ is the change in thickness on application of an electric field[6]. Dividing the maximum strain ($S_{max}$) by the maximum electric field strength ($E_{max}$) gives the piezoelectric coefficient, denoted either as a large signal piezoelectric coefficient, $d_{33}^*$, or a small signal piezoelectric coefficient, $d_{33}$, respectively depending on whether a high or low driving electric field is used[7]. At low electric fields, the strain results principally from the inverse piezoelectric effect, while at high electric fields, there are additional contributions such as from domain switching[8].

The high strain exhibited by numerous piezoelectric materials often necessitates high electric driving fields, risking material breakdown and damage[9]. Applications such as high-power actuators, transducers, and certain medical devices benefit significantly from

high-driven field performance. These applications necessitate robust materials that can withstand and perform efficiently under high-field conditions[10]. Piezoceramics that exhibit high strain under low electric driving fields could offer streamlined control electronics, reduce reliance on high-voltage components, and fundamentally enhance system safety and cost-effectiveness. Precision actuators based on low-field driven high-strain materials would be inherently safer and have higher energy efficiency, thereby prolonging the device lifespan and mitigating the risk of electrical breakdown of the piezoceramic, important for long-term and high-reliability applications.

While the single crystal composition $PbZn_{0.31}Nb_{0.61}Ti_{0.08}O_3$ (PZNTO(SC)) currently stands at the forefront of strain performance, achieving a remarkable 1.7% strain, this is accomplished under a high electric field of 12 kV mm$^{-1}$[11]. In comparison, the highest strain observed in a piezoelectric ceramic, $Pb_{0.45}Bi_{0.375}La_{0.165}Fe_{0.55}Ti_{0.45}O_3$ (PBLFTO), is 1.3% under an electric field of 8 kV mm$^{-1}$[12]. Despite the relatively high strain exhibited by these systems, they face limitations in practical applications due to the proximity of these high electric fields to the electrical breakdown strengths of these materials as well

[1]School of Engineering and Materials Science, Queen Mary University of London, London, UK. [2]STFC ISIS Facility, Rutherford Appleton Laboratory, Oxfordshire, UK. [3]School of Electronic Engineering and Computer Science, Queen Mary University of London, London, UK. [4]Department of Chemistry, Queen Mary University of London, London, UK. ✉e-mail: y.hao@qmul.ac.uk; i.abrahams@qmul.ac.uk; h.x.yan@qmul.ac.uk; l.su@qmul.ac.uk

as the costs linked with high voltage operation. In the context of actuator applications, measurements of strain performed under a relatively low electric field of approximately 2 kV mm$^{-1}$, akin to the driving field in commercial actuators[13], reveal strain values in PZNTO(SC) and PBLFTO that decrease to approximately 0.75% and 0.4%, respectively. These reductions correspond to $d_{33}^*$ values of approximately 3750 and 2000 pm V$^{-1}$, respectively (estimated from the data presented in references[11] and[12]), underscoring the importance of evaluating piezoelectric performance under practical conditions. Despite the impressive performance of PZNTO(SC), its practical application remains limited by its monocrystalline state. The fabrication of single crystal materials necessitates precise control over parameters, an extended crystal growth period, and the utilisation of expensive equipment, rendering them less appealing for practical applications. Moreover, the anisotropic characteristics of single crystals often require sophisticated cutting techniques to obtain samples with desired properties. In contrast, polycrystalline ceramics, fabricated with relative ease and at low cost, exhibit isotropic properties, presenting a more practical alternative.

Numerous approaches have been explored to enhance the piezoelectric performance of ceramics, including aligning of dipoles through strong electric fields[14], controlling the crystallographic orientation in textured ceramics[15], introducing dopants or additives to modify the crystal structure and enhance domain configuration[16,17], optimising grain sizes[18] and post thermal treatment[19]. Despite these efforts, the quest for practical, efficient, and cost-effective piezoelectric materials persists. Currently, lead titanate-based relaxor ferroelectric ceramics, such as Pb(Mg$_{0.33}$Nb$_{0.67}$)$_x$Ti$_{(1-x)}$O$_3$ (PMNPT) at compositions near the morphotropic phase boundary (MPB)[20], represent the state-of-the-art materials, with piezoelectric performance, surpassing traditional commercial PbZr$_x$Ti$_{1-x}$O$_3$ (PZT) based materials. The coexistence of phases at the MPB increases polarisability and improves domain wall movement, allowing for enhancement of electric field induced strain[21].

Rare earth doping has been found to increase piezoelectric performance in PMNPT, especially near the MPB, for example samarium-doped PMNPT exhibits a super high $d_{33}$ value in both ceramic[16] and single crystal states[22]. Similarly, super high $d_{33}$ values have been reported in the case of erbium-doped PMNPT single crystal, which were attributed to an increased ease of polarisation rotation and enhanced domain mobility[23]. Here, we show how careful tuning of the composition and thickness of erbium-doped PMNPT ceramics, enables record high strain values to be achieved at relatively low applied fields. Moreover, the associated mechanism is presented and provides guidance for the development of high strain ferroelectrics. Such materials have the potential to bridge the gap between high strain capability and practical requirements, with implications for ubiquitous piezoelectric devices in diverse applications.

## Results
### Structural and dielectric properties of erbium doped PMNPT ceramics

Ceramics of composition (Er$_{0.025}$Pb$_{0.9625}$(Mg$_{1/3}$Nb$_{2/3}$)$_{1-x}$Ti$_x$O$_3$, ($x$ = 0.26, 0.28, 0.30, 0.32 and 0.34) were prepared by conventional solid-state methods. Although often difficult to resolve, two polar phases, orthorhombic (O phase, in space group $Amm2$) and tetragonal (T phase, in space group $P4mm$), are known to occur at the MPB in similar PMNPT based systems[16]. Close inspection of the data in Fig. 1a reveals subtle compositional changes, with the area around 45° 2θ, where the {200} peak in the pseudo-cubic perovskite (pc) subcell occurs, showing a gradual evolution from a single peak for $x \le 0.30$ compositions to split peaks for $x$ = 0.32 and 0.34 compositions. For compositions in the range $0.26 \le x \le 0.30$, R-factors for the fits to the X-ray powder diffraction (XRD) data in orthorhombic symmetry were marginally better than those in the tetragonal model, while for the $x$ = 0.32 and 0.34 compositions, the best fits obtained were for a mixture of tetragonal and orthorhombic phases (Supplementary Fig. 1), with the former becoming increasingly dominant with increasing $x$-value. This suggests that the MPB lies close to the $x$ = 0.30 composition. The cross-section morphology of the ceramics exhibits closely packed grains with grain sizes ranging from 5 to 15 μm (Supplementary Fig. 2). The neutron diffraction pattern of the $x$ = 0.30 composition at room temperature

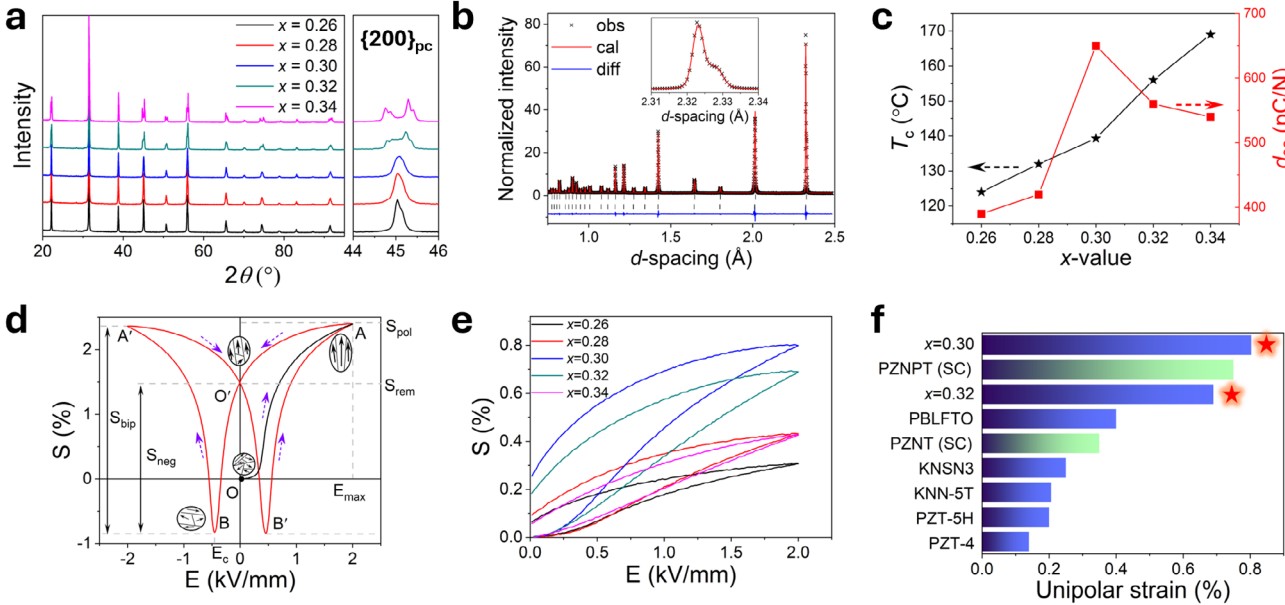

**Fig. 1 | Structure and piezoelectric properties of studied compositions in the Er$_{0.025}$Pb$_{0.9625}$(Mg$_{0.33}$Nb$_{0.67}$)$_{1-x}$Ti$_x$O$_3$ system. a** XRD patterns of studied compositions, with detail of the range between 44 and 46° on the right. **b** Fitted neutron diffraction profile for the $x$ = 0.30 composition at room temperature. **c** Compositional dependence of Curie temperature and Piezoelectric coefficient.

**d** Bipolar *S-E* loops for the $x$ = 0.30 composition under an electric field of 2 kV mm$^{-1}$. **e** Unipolar strain of poled ceramic samples of all studied compositions. **f** Comparison of unipolar strain at 2 kV mm$^{-1}$ observed in other high-strain materials (KNSN3: K$_{0.485}$Na$_{0.485}$Sr$_{0.03}$NbO$_3$, KNN-5T: Textured (K,Na)NbO$_3$ based ceramic)[11,12,37,38] with that found in the present study.

was fitted using a model consisting of a mixture of O and T phases, accounting accurately for the $\{112\}_{pc}$ peaks at $d$-spacings between 2.31 and 2.34 Å (Fig. 1b). Crystal and refinement parameters are summarised in Tables S3–S4, revealing refined weight fractions of 71.8(4)% and 28.2(5)% for the T and O phases, respectively.

Despite PMNPT based materials being classified as relaxor ferroelectrics[16], the dielectric permittivity and loss of the studied compositions exhibit minimal frequency dependence (Supplementary Fig. 3). The Curie temperature ($T_c$) is found to increase with increasing titanium concentration (Fig. 1c), with the dielectric loss higher below $T_c$ than above it, attributable to domain wall friction[24,25]. The compositional variation of $d_{33}$ reveals it to increase with increasing $x$-value to a maximum of $650\,pC\,N^{-1}$ for the $x = 0.30$ composition before decreasing at higher $x$-values. This maximum in $d_{33}$ coincides with the onset of the structural transition from a single orthorhombic phase to a mixture of orthorhombic and tetragonal phases, another indicator of the close proximity of this composition to the MPB.

The bipolar $P$-$E$ and $S$-$E$ plots for the $x = 0.30$ composition display a symmetrical butterfly shape (Fig. 1d). Starting from an unpoled sample (point O), application of a positive electric field induces a strain including electrostriction, piezoelectric, and non-180° domain switching contributions. In the case of Er-doped PMNPT, there is also a contribution from a field induced phase transition. The maximum strain is achieved at the maximum applied field (point A). On removal of the applied field, the electrostriction and piezoelectric contributions fall to zero, but not all the strain from the field induced phase transition and non-180° domain switching is recovered (point O′). The difference between O and O′ represents the remanent strain ($S_{rem}$) and for the $x = 0.30$ composition at 20 °C $S_{rem}$ has a value of 1.54%. Point O′ represents the starting position for poled samples and is taken as the zero strain position for bipolar measurements. Thus, negative strain ($S_{neg}$) and positive strain ($S_{pos}$) are defined as the difference between the strain at zero electric field (point O′) and the minimum and maximum strains during bipolar cycles, respectively, while the bipolar strain ($S_{bip}$) is the difference between minimum and maximum strain values. In the present case, a remarkably high value of 3.19% is obtained for bipolar strain in the $x = 0.30$ composition at a maximum field of $2\,kV\,mm^{-1}$, corresponding to a $d_{33}^*$ value of 15950 pm V$^{-1}$.

Typically, in device applications, a unipolar driving field is used and hence unipolar strain ($S_{uni}$) measurements on poled samples are more reflective of the behaviour of the material under operating conditions. Unipolar $S$-$E$ loops for ceramic samples previously poled at $2\,kV\,mm^{-1}$ are shown in Fig. 1e. $S_{uni}$ is found to increase from 0.31% for the $x = 0.26$ composition to a maximum of 0.80% for the $x = 0.30$ composition, corresponding to a $d_{33}^*$ value of 4014 pm V$^{-1}$, before decreasing at higher values of $x$. It is noteworthy that $S_{uni}$ for the $x = 0.30$ composition increases relatively slowly at low electric fields, consistent with relatively low $d_{33}$ values. However, when the electric field exceeds a certain value (ca. $0.25\,kV\,mm^{-1}$), the strain exhibits a sharp increase, attributed to the field-induced phase transition from tetragonal to orthorhombic structures. As the electric field continues to increase, the strain appears to approach a saturation point, with a slower rate of increase. The $S_{uni}$ value of 0.8% at $2\,kV\,mm^{-1}$ for the $x = 0.30$ composition represents the highest observed value for any piezoelectric material and is approximately twice that of the leading lead-based PBLFTO ceramic (Fig. 1f).

## Electric field, sample thickness and thermal dependence of piezoelectric performance

Figure 2a shows the change in $S_{bip}$ as a function of electric field for the studied compositions with details of $I$-$E$, $P$-$E$ and $S$-$E$ loops given in Supplementary Fig. 4. The $x = 0.30$ composition exhibits a saturation polarisation of $0.33\,C\,m^{-2}$ and remanent polarisation of $0.29\,C\,m^{-2}$, with a coercive field of $0.45\,kV\,mm^{-1}$, indicating that only a small fraction of dipoles reorientate after removal of the electric field,

consistent with the high remanent strain seen at 20 °C. Although the bipolar strain of the studied compositions increases with increasing electric field, the rate of this increase slows, such that it reaches a near steady state at around $2\,kV\,mm^{-1}$. For the $x = 0.30$ composition, the bipolar strain remains high (>3%) for all electric fields above $1\,kV\,mm^{-1}$. In contrast, $d_{33}^*$ decreases with increasing field strength, with a maximum value of $2.7 \times 10^4\,pm\,V^{-1}$ obtained at $1\,kV\,mm^{-1}$ for the $x = 0.30$ composition (Supplementary Fig. 5). This value represents the highest $d_{33}^*$ value under bipolar conditions ever reported, an order of magnitude higher than those of most piezoceramics. The variation of $S_{uni}$ with applied field (Fig. 2b, with details of unipolar $S$-$E$ loops in Supplementary Fig. 6) reflects that seen for $S_{bip}$, increasing with increasing field, but with the rate of increase diminishing with increasing field. This behaviour is primarily due to the reorientation of domains. As the electric field increases, the domains gradually align, approaching a fully aligned state. Once this alignment nears completion, the rate of strain increase diminishes, leading to a decrease in $d_{33}^*$ with further increase in electric field[11,26]. The $x = 0.30$ composition showed the highest $S_{uni}$ value of 0.9% at $5\,kV\,mm^{-1}$ and the highest $d_{33}^*$ value of 6437 pm V$^{-1}$ at $1\,kV\,mm^{-1}$. Previous studies on piezoelectric materials have often focused on enhancing the maximum strain value but typically at the cost of requiring high electric fields, resulting in relatively low $d_{33}^*$ values (Fig. 2c). In contrast, the $x = 0.30$ and 0.32 compositions reported here achieve high strain levels at a relatively low electric field of $1\,kV\,mm^{-1}$. Specifically, the $d_{33}^*$ and strain values for unipolar measurements for the $x = 0.32$ and 0.30 compositions far exceed those of previously studied materials and set a benchmark for high-strain materials.

To investigate the thermal stability of strain behaviour in this system, bipolar $S$-$E$ loops for the $x = 0.30$ composition were measured from room temperature up to 140 °C (Fig. 2d). Minimal strain variation occurred up to 75 °C, but above this temperature a dramatic decrease in bipolar strain was observed. This is attributed to a decrease in distortion with increasing temperature in both T and O phases accompanied by the gradual transition from orthorhombic to tetragonal symmetry (see below)[27]. The electric field-induced unipolar and bipolar strain exhibits minimal variation over 1000 cycles, demonstrating excellent stability (Supplementary Fig. 7).

In addition to thermal and field effects on strain, the effect of sample thickness was investigated for the $x = 0.30$ composition (Fig. 2d–f). A clear thickness dependence of electric field-induced strain is observed, with strain values increasing from 0.39% at a thickness of 0.92 mm to 2.18% at 0.30 mm. However, this does not appear to be related to changes in polarisation with no significant changes in saturation polarisation with thickness (Fig. 2d). It has been proposed that in ferroelectric systems under an applied field there is an inhomogeneous field distribution, arising from lower permittivity of the surface layers compared to the bulk material[28]. While this could explain the observation of increasing strain with decreasing sample thickness (Fig. 2f), one would also expect that as sample thickness decreases overall permittivity would also decrease as the surface to bulk ratio increases. However, as demonstrated in Supplementary Fig. 8, above a thickness of 0.34 mm there is little change in permittivity and loss with increasing thickness. Indeed, permittivity and loss are seen to increase with decreasing thickness below 0.34 mm. An alternate explanation for the observed thickness dependence of strain is to consider the surface layer as offering reduced mechanical constraints to high strain allowing for easier accommodation of field induced distortion. Since the strain of the sample is a combination of bulk and surface strains, samples with a higher surface to bulk ratio would be expected to exhibit higher values of total strain as is observed. For samples with thicknesses greater than 0.58 mm, similar negative and positive values of displacement ($\Delta l$, where $\Delta l$ = strain × sample thickness) are observed under applied electric field (Supplementary Fig. 9). The displacement loops are symmetrical, in contrast

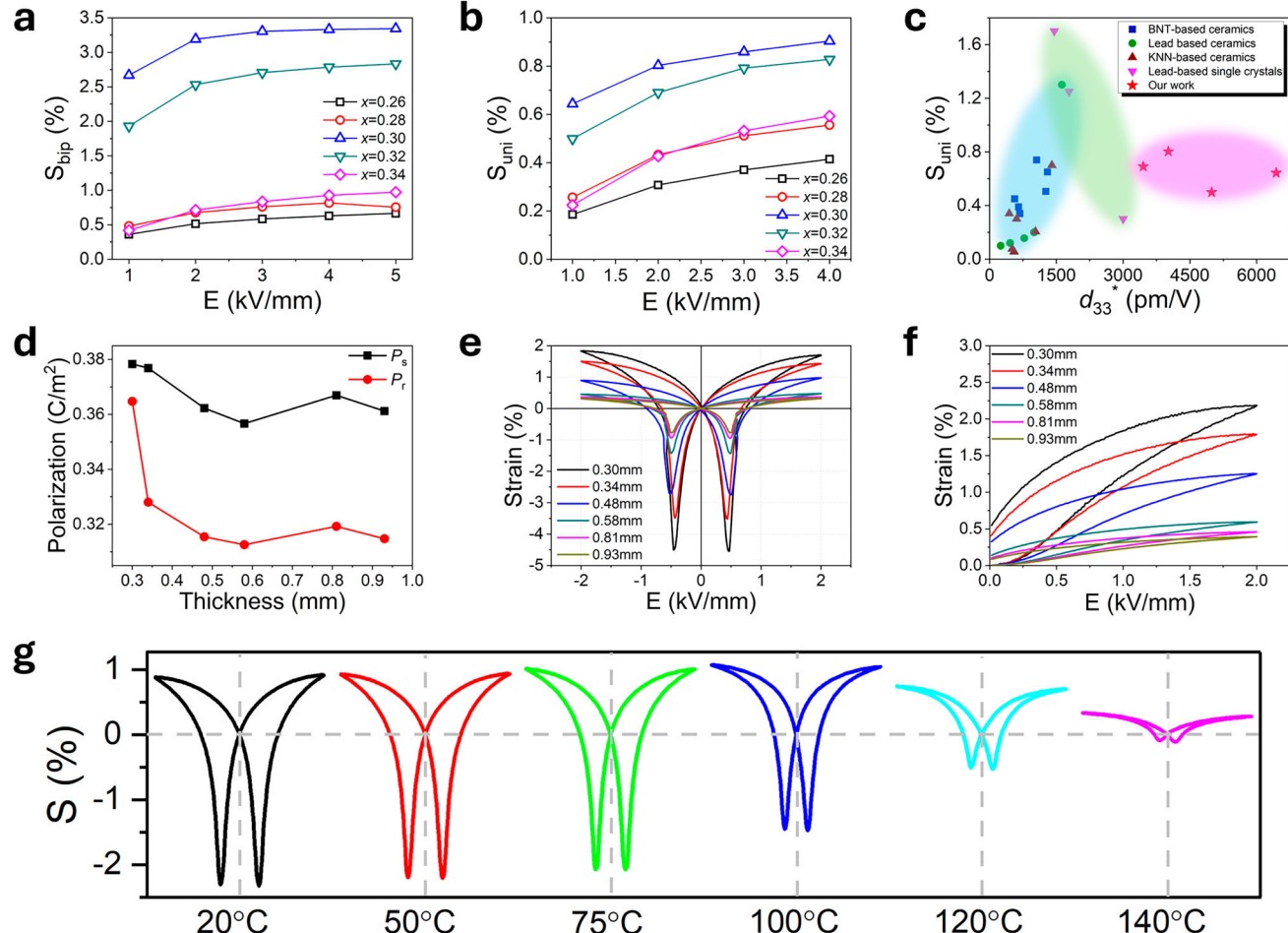

**Fig. 2 | Electric induced strain performance of compositions in the Er$_{0.025}$Pb$_{0.9625}$(Mg$_{0.33}$Nb$_{0.67}$)$_{1-x}$Ti$_x$O$_3$ system.** Electric field dependence of (**a**) bipolar and (**b**) unipolar strain for studied compositions at room temperature. **c** Comparison of unipolar strain and $d_{33}^*$ values in other high-strain materials[22,37–49] with those in the present study. Electric field induced (**d**) polarisation, (**e**) bipolar and (**f**) unipolar strain at room temperature for samples of the $x = 0.30$ composition of different thickness. **g** Bipolar S-E loops of the $x = 0.30$ composition measured under 2 kV mm$^{-1}$ at selected temperatures.

to the work of He et al.[29], where asymmetry in the loops was attributed to bending deformation, suggesting that in the present case bending deformation is not a significant contribution to the observed high strain. For samples thinner than 0.58 mm, the displacement is significantly higher, suggesting the surface layer effect is dominant in thinner samples. Micro-indentation tests further support this hypothesis, showing that the hardness of the samples decreases with decreasing thickness, indicating a lower Young's modulus (Supplementary Fig. 10)[30]. The reduction in Young's modulus lowers the mechanical clamping effect, making thinner samples more susceptible to deformation under the applied electric field.

**Structural origin of high strain in the erbium doped PMNPT system**

The thermal evolution of the neutron diffraction patterns for the $x = 0.30$ composition shows three distinct regions, viz.: below 7 °C, between 7 and 140 °C, and above 140 °C (Fig. 3a). Notably, the diffraction patterns change rapidly between 7 and 140 °C, indicative of significant structural changes. Rietveld refinement reveals that on heating from −173 to 7 °C a gradual decrease in the O phase fraction and an increase in that for the T phase occurs (Fig. 3b). Further heating leads to a significant decrease in the O phase fraction which eventually disappears at 80 °C, leaving the T phase (79.8%), along with a small amount of the cubic (C) phase (20.2%). At temperatures between 80

and 140 °C, the T phase fraction rapidly falls to leave only the C phase at 140 °C.

Spontaneous polarisation of both O and T phases generally decreases with increasing temperature[31], with the O phase exhibiting slightly higher values (Fig. 3c). The total polarisation shows a gradual decrease on heating and drops sharply above 80 °C. The saturation polarisation and remanent polarisation extracted from *P-E* measurements (Supplementary Fig. 11), exhibit a similar decreasing trend upon heating at low temperatures, with a rapid drop in remanent polarisation above 75 °C (Supplementary Fig. 12) due to the O → T phase transition. At temperatures above 150 °C, the current peak at around zero field becomes increasingly broad, which is characteristic of a relaxor ferroelectric and is often described by the existence of polar nanoregions (PNRs) within the material, the concentration of which decreases with increasing temperature[31]. A comparison of the XRD patterns for the $x = 0.32$ and 0.34 ceramic pellets before and after poling reveals changes in the intensities of the peaks associated with the O and T phases (Supplementary Fig. 13), indicating that the poling process induces a partial irreversible phase transition from T to O phases. Compared to that of the unpoled state of the $x = 0.30$ composition, the dielectric spectrum of the poled sample exhibits an additional dielectric permittivity anomaly and a dielectric loss peak at *ca.* 75 °C (Supplementary Fig. 14), consistent with the observed strain degradation at temperatures over 75 °C (Fig. 2d), indicating the

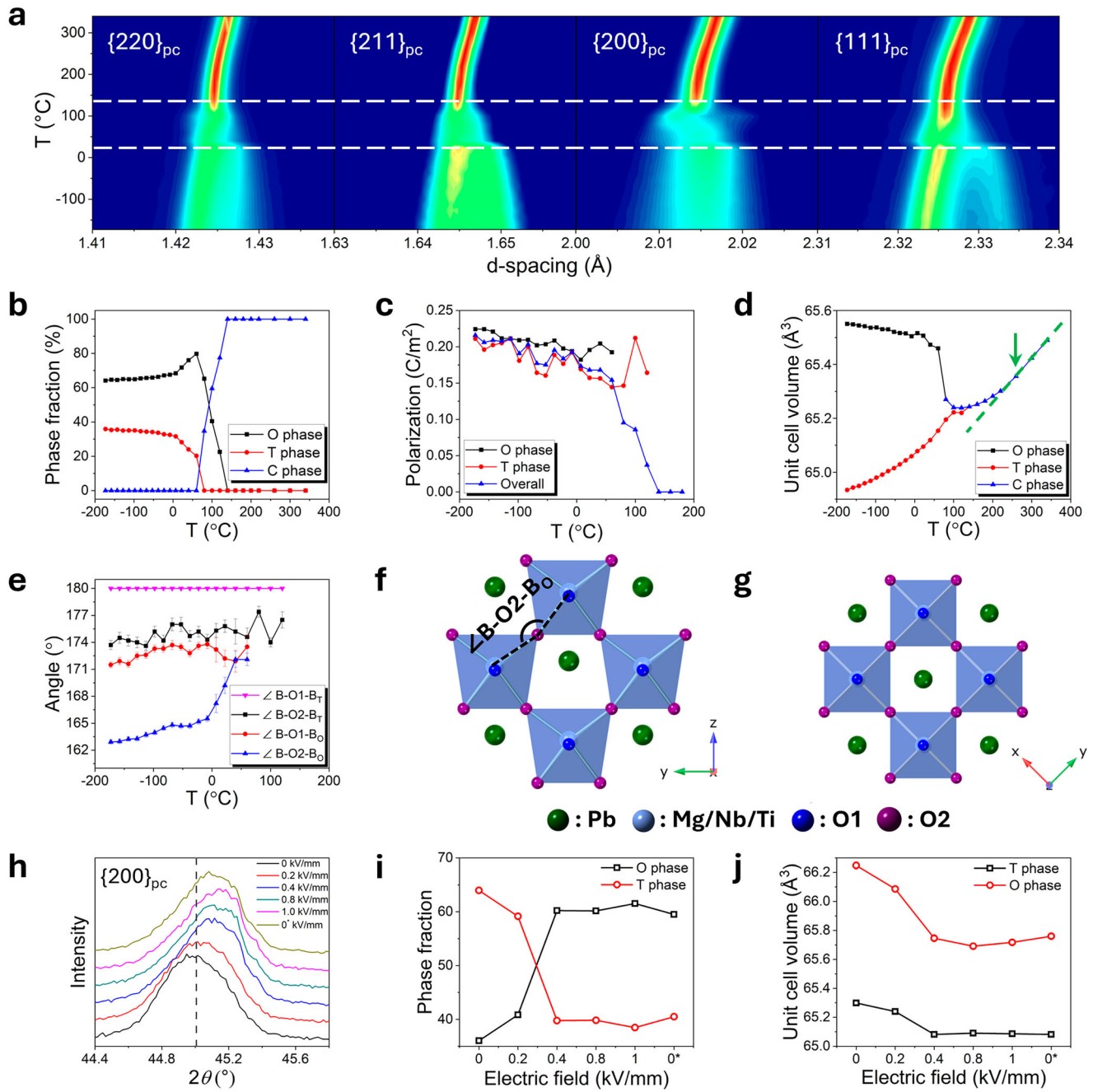

**Fig. 3 | High resolution neutron diffraction study of Er$_{0.025}$Pb$_{0.9625}$(Mg$_{0.33}$Nb$_{0.67}$)$_{0.7}$Ti$_{0.3}$O$_3$. a** Contour plot showing thermal variation of selected neutron diffraction peaks. Temperature dependencies of (**b**) phase fraction, (**c**) polarisation, (**d**) unit cell volume and (**e**) ∠B-O-B bond angle with error bars (standard error) between B-site cations and oxygen for O, T and C phases of the x = 0.30 composition. Crystal structures of (**f**) O phase and (**g**) T phase (images correspond to the x = 0.30 composition at −173 °C). Electric field dependencies of (**h**) XRD patterns, (**i**) phase fraction and (**j**) pseudo-cubic unit cell volume. 0* is zero field after removal of applied electric field.

additional features in the permittivity and loss spectra are attributable to a field induced phase transition between O and T phases.

To allow comparison of phases, the unit cell volume of the orthorhombic phase (O) was converted into that of the pseudo-cubic (pc) subcell, $V_{pc} = 0.5 × V_O$. Interestingly, the pseudo cubic unit cell volume of the O phase shows a gradual decrease on heating, which contrasts with the expansion observed in the T and C phases (Fig. 3d). This negative thermal expansion of the O phase is associated with a decrease in distortion, most evident in the bond angle between B-site octahedra (∠B-O-B$_O$), with the ∠B-O2-B$_O$ angle increasing steadily from 162.9° at −173 °C to 165.5° at −8 °C, before increasing sharply to

172.1° at 40 °C, virtually equal to the ∠B-O1-B$_O$ angle as the orthorhombic phase approaches tetragonal symmetry (Fig. 3e). The linear thermal expansion in the C phase above 260 °C is consistent with the Burns temperature of *ca.* 270 °C (Supplementary Fig. 15), marking the transition to the fully paraelectric cubic phase.

It is widely accepted that superior piezoelectric properties originate from the MPB regions, attributable to their varied spontaneous polarisation directions, yet a detailed explanation of this phenomenon is still lacking. Here, considering the accompanying variable temperature strain behaviour (Fig. 2d) and detailed structure analysis, we propose that the exceptionally high strain observed in the x = 0.30

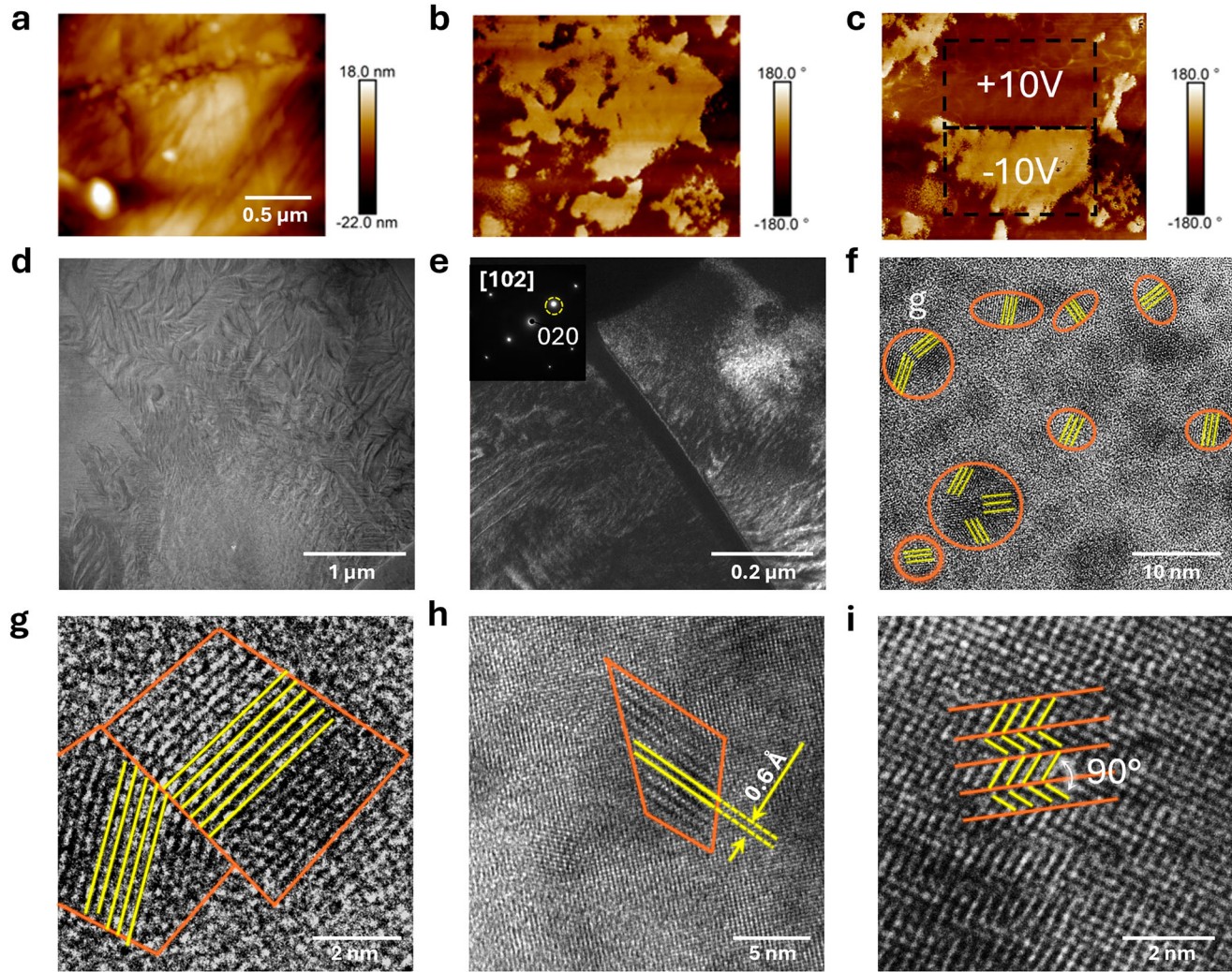

**Fig. 4 | PFM images of $x$ = 0.30 composition in the $Er_{0.025}Pb_{0.9625}(Mg_{0.33}$ $Nb_{0.67})_{1-x}Ti_xO_3$ system. a** morphology and (**b**, **c**) phase images before and after voltage application, along with (**d**) bright, (**e**) dark (using 020 spot) and (**f–i**) high resolution TEM images. The regions outlined by orange lines and the solid yellow lines/arrows in (**f–h**) represent the polar regions and atomic alignment, respectively.

composition from room temperature to 75 °C at low applied fields is not solely due to the mixture of O and T phases at the MPB. Rather, it is the large difference in distortion between the two phases which leads to a large difference in the pseudo-cubic unit cell volume and high strain in the lower symmetry phase, evident in the distortion of the ∠B-O2-$B_O$ angle (Fig. 3e); while the relative stability of these phases allows them to readily transition between each other at relatively low applied fields. Figure 3h–j and Supplementary Fig. 16. summarise the results of an XRD study of the $x$ = 0.30 composition carried out under different electric fields. The $\{111\}_{pc}$ and $\{200\}_{pc}$ diffraction peaks exhibit a noticeable shift toward higher 2θ angles as the electric field increases. When the electric field reaches 0.4 kV mm⁻¹, there is a rapid decrease in the T phase fraction and a corresponding increase in the O phase fraction, indicating an electric field induced phase transition from T to O phases. There is little further increase in the O phase fraction with further increase in electric field. After removal of the electric field, the T phase exhibits a small recovery in its phase fraction with respect to the O phase. The unit cell parameters and unit cell volume show a clear decrease with electric field up to 0.4 kV mm⁻¹, which is close to the coercive field (ca. 0.45 kV mm⁻¹). A decrease in volume is consistent with thermodynamic considerations as the disordered higher symmetry T phase transforms to the ordered lower symmetry O phase. With further increases in the electric field, the cell parameters show

only a slight decrease. After removing the electric field, the poled sample did not fully recover to its original state, retaining reduced cell parameters. This is consistent with the P-E data, which show only a slight decrease from saturation polarisation to remanent polarisation upon removal of the electric field.

**Ferroelectric domain and PNR characterisation**

Piezoresponse force microscopy (PFM) was carried out to examine the local polar structure in the $x$ = 0.30 composition (Fig. 4). The phase images reveal different domain sizes, ranging from nanometres to micrometres, randomly distributed in the materials. After application of ±10 V DC bias fields on the sample, dark and bright regions were observed in the phase image, reflecting the polar domain structure. The change in amplitude and phase under an electric field was measured at a single point and is shown in (Supplementary Fig. 17). Two valley points in amplitude were found at ca. ±2 V corresponding to domain switching. Raman spectroscopy was used to analyse the local structure in both unpoled and poled ceramic samples (Supplementary Fig. 18). The peak, corresponding to the $A_1$ mode, exhibits a clear increase in intensity after poling (Supplementary Fig. 19). The difference in peak positions between $C_3$ and $C_2$ modes decreases after poling (Supplementary Fig. 20), indicating the gradual merging of these peaks. While the relative intensity ratio $C_3/C_2$ for the unpoled samples

exceeds one, it decreases to less than one after poling. The observed changes are similar to those seen in perovskite structured $Bi_{0.5}Na_{0.5}TiO_3$ based ferroelectrics, where local structural changes are caused by an electric field induced phase transition from tetragonal to orthorhombic structures[32].

Energy-Dispersive X-ray spectroscopy images confirm the homogeneity of the elemental distribution in the system (Supplementary Fig. 21). Selected area electron diffraction (SAED) was performed on the $x = 0.30$ composition to investigate the local structure within the system. All the observed electron diffraction spots can be indexed based on the orthorhombic structure along [100], [010], [103] and [221] directions (Supplementary Fig. 22). However, the SAED images do not easily enable distinction between the orthorhombic and tetragonal structures owing to their similarities. A transmission electron microscopy (TEM) image of the $x = 0.30$ composition reveals a lamellar domain structure as well as complicated domain configurations (Fig. 4e) and (Supplementary Fig. 23a). At higher resolution, the TEM image shows numerous ellipsoidal polar nano regions, ~3–5 nm in size, each exhibiting varying contrast and locally ordered arrangements, randomly distributed within a single grain (Fig. 4f and Supplementary Fig. 23b). Close inspection of region G in Fig. 4f identified two distinct ordered arrangements forming a twin boundary (Fig. 4g). Additionally, ordered stripes with thickness of ~6 Å (Fig. 4h) and 90° twin boundaries (Fig. 4i) were observed locally. The presence of twin structures will contribute to the ease of switching at low field and high permittivity and will thus have some contribution to strain.

### Origins of high strain in the Er-PMNPT system

A number of factors appear to contribute to the observation of high strain at low fields in the Er-PMNPT system. Firstly, the fact that Er-PMNPT is a relaxor ferroelectric, characterised by polar nanoregions makes it particularly responsive to low applied fields. Indeed, most reported high-strain materials are relaxor ferroelectrics, for example systems based on bismuth sodium titanate[33] and bismuth ferrite[12] as well as other PMNPT[16] based systems. Secondly, the accessibility of a field induced phase transition at room temperature and low applied fields is an important factor. The occurrence of such a phase transition, at which the free energies of the two polymorphs involved are equal, is dependent on composition, temperature and electric field. Thirdly, while the two polymorphs have equal free energies at the phase transition, they have significantly different polarisation vectors. In the tetragonal phase, the polarisation vector lies parallel to the $c$-axis of the tetragonal cell equivalent to <100> in the pseudo-cubic perovskite cell. In the case of the orthorhombic phase, the polarisation vector is rotated through 45° to the <110>$_{pc}$ direction. This rotation causes significant strain and leads to structural distortion which manifests itself as the phase transition to the orthorhombic phase. The in-situ XRD results support this hypothesis, demonstrating that as the electric field increases, the material undergoes structural distortion leading to an increase in the phase fraction of the orthorhombic phase. Finally, as discussed above, the thickness of the sample has a significant influence on strain, with thinner samples showing an increase in strain with decreasing thickness associated with a lowering of Young's modulus, making thinner samples more susceptible to deformation and accommodation of high strain

## Outlook

$Er_{0.025}Pb_{0.9625}(Mg_{1/3}Nb_{2/3})_{1-x}Ti_xO_3$ shows exceptionally high electric field induced strain, with the $x = 0.30$ composition achieving an extraordinarily high $d_{33}^*$ value, establishing a benchmark in this field. The ultrahigh strain seen in the present system is attributed to a combination of factors, including the sensitivity of the polar nanoregions in this relaxor ferroelectric to low applied fields, the proximity of the system to the MPB and the small difference in the relative stabilities of the O and T phases which enable the transition at low applied

fields, the change in polarisation vector between the T and O phases and the accompanying structural distortion on application of the applied electric field, and finally the thickness of the sample, with thin samples more ready to accommodate high strain. Consequently, this material achieves ultrahigh strain at relatively low electric field strengths, further enhancing its attractiveness for practical applications. This remarkable phenomenon merits further theoretical studies to clarify the physical mechanism behind the observed high strain.

## Methods

### Sample preparation

Ceramics of general formula of $Er_{0.025}Pb_{0.9625}(Mg_{0.33}Nb_{0.67})_{1-x}Ti_xO_3$ ($x = 0.26$, 0.28, 0.30, 0.32 and 0.34) were prepared by a two-step solid state method[34]. In the first step, stoichiometric amounts of magnesium oxide (MgO, 99%, Sigma Aldrich) and niobium oxide ($Nb_2O_5$, 99.9%, Sigma Aldrich) were ball milled together in ethanol at 180 rpm for 20 h using a planetary ball mill (Fritsch Pulverisette 5/4 classic line, Germany) with zirconia balls. After drying, the powder was calcined in an alumina crucible at 1200 °C for 2 h to form $MgNb_2O_6$ (confirmed by X-ray powder diffraction). In the second step, erbium oxide ($Er_2O_3$, 99.9%, Sigma Aldrich), lead oxide ($Pb_3O_4$, 99%, Sigma Aldrich), titanium oxide ($TiO_2$, 99.8%, Sigma Aldrich) and $MgNb_2O_6$ were similarly ball milled together in ethanol at 180 rpm for 20 h. The mixture slurry was dried and sieved through a 250 μm sieve followed by calcination at 840 °C for 4 h. After cooling, the calcined powder was ball milled again, dried and sieved. The resulting powder was pressed uniaxially into 13 mm diameter pellets of ~1–2 mm thickness at a pressure of 150 MPa. Sintering of the pellets was carried out at 1200 °C for 2 h. Parallel syntheses were carried out and confirmed the reproducibility of the results.

### Materials characterisation

A PANalytical X'Pert Pro diffractometer, fitted with an X'Celerator detector was used to collect X-ray powder diffraction (XRD) data with Ni-filtered Cu-Kα radiation (λ = 1.5418 Å), over the 2θ range 5–120°, with a step width of 0.0334° and an effective count time of 50 s per step. For in situ XRD measurements under applied field, silver paste was applied to the surface of a sample of the $x = 0.30$ ceramic with a 5 mm gap between electrodes exposed for illumination by the X-ray beam (Supplementary Fig. 24a). Cu wires were used to connect the electrodes to a 3B Scientific power supply. Measurements were recorded at applied fields from 0 to 1 kV.mm⁻¹. Neutron diffraction data were collected using the high-resolution powder diffractometer (HRPD) at the ISIS Facility, Rutherford Appleton Laboratory, UK. Measurements were conducted from −173 to 22 °C and 40 to 340 °C, with the sample contained in a $15 \times 20 \times 10$ mm³ vanadium slab can and 8 mm diameter vanadium cylindrical can, respectively. Data sets corresponding to proton beam current equivalents of 70 μA h were collected at −173 °C, 98 °C, −23 °C, 22 °C, 120 °C and 200 °C, with shorter scans of around 35 μA h at other temperatures. Data acquired in the time-of-flight range 32–120 ms on the backscattering (158.46° <2θ< 176.11°) detector bank, corresponding to $d$-spacings between 0.65 and 2.58 Å were used in the subsequent analysis. XRD and neutron patterns were fitted by Rietveld analysis using the GSAS suite of programs[35], with tetragonal ($P4mm$), orthorhombic ($Amm2$) and cubic ($Pm\text{-}3m$) perovskite models[14]. The morphology of the sintered ceramics was examined by scanning electron microscopy (SEM, FEI Inspect-F Oxford). Piezoresponse force microscopy (PFM) was carried out on an AFM system (Bruker Dimension Icon, US) using an SCM-PIT-V2 conductive probe (Bruker, US). A ceramic sample was ground down to $ca$. 140 μm thickness and then ion milled until electron transparency was achieved. Transmission electron microscopy (TEM, JEM-F200, Japan) was used to examine the local structure of the $x = 0.30$ composition. The morphology of the sample surface was examined by scanning electron microscopy (FEI Inspect F SEM). Raman scattering

spectra were acquired at room temperature for unpoled and poled ceramic samples using a Via™ confocal Raman microscope equipped with a 785 nm wavelength laser. Hardness tests were conducted using a Mitutoyo Vickers Hardness Testing Machine (HM200 series). Each sample underwent seven indentations under a 100 g load, applied with a HM−210 Vickers diamond indenter.

## Dielectric and piezoelectric measurements

For electrical measurements, silver paste (Gwent Electronic Materials Ltd. Pontypool, UK.) was applied to parallel surfaces of ceramic pellets and pellets heated to 450 °C for 10 min to form the electrodes. Dielectric measurements were carried out using a precision impedance analyser (Agilent 4294) at room temperature and an LCR meter (Agilent 4284) connected to a furnace for variable temperature measurements. Ferroelectric measurements such as current density-electric field ($I$-$E$), polarisation-electric field ($P$-$E$) and strain-electric field ($S$-$E$) loops were measured using a ferroelectric hysteresis tester (NPL, UK). Ceramic samples were placed between a spherical top electrode and a flat bottom electrode (Supplementary Fig. 24b) and the electric voltage was applied in a triangular waveform at 1 Hz[36]. The piezoelectric coefficient ($d_{33}$) was measured using a quasi-static $d_{33}$ meter (ZJ-3B Institute of Acoustics Academia Sinica, China). The poling process was carried out by applying an AC electric field of 5 kV mm$^{-1}$.

## Data availability

The data supporting the findings of this study are available in the paper and the Supplementary Information. The neutron data used in this study are available at https://doi.org/10.5286/ISIS.E.RB2310444. Any other relevant data are also available from the corresponding authors upon request.

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

## Acknowledgements

Dr Chris Howard at the ISIS Facility, Rutherford Appleton Laboratory is thanked for his help in neutron data collection. Dr Richard Whitely and Dr Subash Rai at Queen Mary University of London are thanked for their help in X-ray and electron microscopy data collection, respectively. This work was funded by the Engineering and Physical Sciences Research Council (EPSRC) (grant no. EP/W004399/1) and Kidney Research UK; EPSRC (grant no. EP/R035393/1, EP/X02542X/1). The Science and Technology Facilities Council is thanked for a neutron beam time award at the ISIS facility, Rutherford Appleton Laboratory (RB2310444).

## Author contributions

H.Z. prepared the materials, conducted the materials characterisation and data analysis. Z.L., Y.W., A.F., and T.S. assisted with data analysis. H.Z., I.A., and H.Y. contributed to the discussion of results. H.Z. wrote the original draft, and all authors contributed to editing the final version of the manuscript. Y.H. and L.S. acquired the funding and supervised the project.

## Competing interests

The authors declare no competing interests.
