## [Transparent Peer Review file · Nature Communications]

Phase transformation in lead titanate based relaxor ferroelectrics with ultra-high strain

Corresponding Author: Dr Isaac Abrahams

Version 0:

Reviewer comments:

Reviewer #1

(Remarks to the Author)

Zhang et al reported the high-strain induced by small electric field in erbium-doped PMN-PT ceramics. The piezoelectric properties of this material are overall good, which exhibit high strain (3.19% bipolar and 0.8% unipolar) under a low applied field of 2 kV/mm, resulting in record-breaking piezoelectric coefficients (d_{33}^* values of 15950 and 4014 pm/V, respectively). Authors proposed a mechanism for such a superior property, namely, these exceptional properties stem from both the morphotropic phase boundary and differential distortion between the two phases present at the boundary.

Although this work reports interesting and great materials performance that is the only selling point for a publication, I have concerns about the interpretation of the experimental results and the discussion of the mechanism, which I think should be further clarified. At current stage, the manuscript should be rejected and the manuscript should be rewritten.

- 1) It is not fair to overly emphasize the importance of low-driven field. Sometimes high-drive applications are also important.
- 2) Lack of experimental evidence for proposed mechanism. Authors performed temperature dependence of structural analysis by neutron diffraction. Yet, I think an electric-field dependent structural analysis is more important here. Authors said there is a differential distortion between the two phases present at the boundary. If they want to prove it, they should conduct in situ electric-field driven XRD or neutron diffraction experiments to observe the structural change under the small and large applied field.
- 3) Furthermore, many ceramics and/single crystals are designed with MPB composition, most of which also have two or more phases. For various phases, the lattice distortion must be different. Authors cannot use such a normal concept to understand their discovery.
- 4) Authors should perform more detailed study on PNR and dielectric properties, also the loss behavior. Current PFM results are not enough. Similar PNR behavior are reported in many previous works.
- 5) The material system reported here is not new. Many works on similar or even same materials and their piezoelectric properties have been reported. It is interesting that why previous works have not observe the superior results reported here. I ask this question because authors said their samples were prepared by "conventional solid-state methods."

Reviewer #2

(Remarks to the Author)

The authors designed a rare earth element doped PMN-PT near MPB ceramic for high-strain applications. Although the presented data are attractive in this field, the reliability of the performances and scientific understandings of the origins of the giant electro-strain need to be reinforced before publication in Nature Communications.

- 1) The first concern focuses on the strain data. Fig. 1e presents severely unrecovered strain in the field on/off cycles, which is unexpected in converse piezoelectric effect, for example, the ref. 9 they cited, showing a normal shape. Is there any explanation for this phenomenon? If the unrecovered strain cannot be used in a practicable application, then the claim they made, i.e. 0.8% strain is not accurate.
- 2) The authors should provide multiple cycle measurements of their samples, both unipolar and bipolar strain, seeing the stability of the performances in cyclic tests.
- 3) Both the bipolar and unipolar strain data are very impressive, I am wondering whether the author can provide the visual

test, for example, a high-speed camera with sufficient resolution since its strain should be sufficiently high for capturing. Is there any thickness-dependent phenomenon, for example, electro-strain increases with reduced thickness?

4) The ultra-high strain data needs more quantitative analysis. According to the authors' description, they attributed the strain to be induced by the phase transition between O and T phases. Phase transition could involve high strain, while they are lack of direct evidence that the transition occurs under electric fields. In-situ XRD measurement under various electric fields can provide proof.

5) On the other hand, the unipolar strain exhibits an ultra-high contraction strain during the domain switch, which is even higher than 3%. I do not think the phase transition can solely be interpreted. The difference in unit cell volume between the O and T phases is lower than 1% (fig, 3d), and the ceramics are untextured. What are the concrete differences in lattice parameters from T to O? The in-plane strains may also help understand the giant performances, I recommend the author to measure them.

6) Fig. 3b, 3c, 3d, and 3e present differences between O and T phases as a function of temperature, indicating the phase transition behaviors. Meanwhile, the author also claims their performance is attributed to the morphotropic phase boundary, which should be nearly independent of temperatures. The MPB can contain multiple energy degenerated phases, but it is very difficult to claim their contents and temperature-dependent concentrations. The authors should not simply attribute the performance to MPB. Such giant strain associated with phase transition is more like a critical phenomenon in ferroelectrics rather than MPB (Nature, 2006, 441(7096): 956-959).

7) The authors are expected to provide theoretical calculations, such as phase field simulation or density functional theory, to support the claim that the ultra-high strain comes from lattice strain due to phase transitions.

Reviewer #3

(Remarks to the Author)

The authors achieve attractive piezoelectric properties in the erbium-doped lead magnesium niobium titanate ceramics. However, some important issues have yet been explained.

1. Compared with the extremely high d_{33}^* and unipolar strain (under 2 kV/mm), the d_{33} value is quite low. The authors should explain it clearly.

2. How about the leakage behavior of the samples?

3. The authors should compare the dielectric properties, including both the dielectric constant and dielectric loss for the samples under various frequency and temperature.

4. The authors report that the d_{33}^* decreases with increasing the field strength. Why does this happen?

5. The authors report that for the sample 0.30, the O phase could easily transform into T phase. Besides the evidence in distortion of the B-O₂-Bo angle, is there any other evidences? Maybe, some simulation results should be added. Another question is whether it is easy to fabricate the sample 0.30 with so many metastable phases.

6. For the PFM analyses, the contrast of the "-10 V/10 V" area is not 180/-180. Why does this happen? Is 10 V enough for poling the samples?

7. In the experimental section, the poling information should be provided.

Reviewer #4

(Remarks to the Author)

I co-reviewed this manuscript with one of the reviewers who provided the listed report. One of the reviewers who provided the listed reports.

Version 1:

Reviewer comments:

Reviewer #1

(Remarks to the Author)

The authors have addressed my concerns. It can be published in Nature Communications.

Reviewer #2

(Remarks to the Author)

The authors have partially addressed my concerns. However, I remain skeptical about the manuscript due to the lack of convincing answers to my major comments.

1) For ceramics, a strain exceeding 3% is likely to cause material fracture and failure unless it originates from a phase transformation (not an elastic process). However, the existing XRD evidence does not support such a significant strain. On the other hand, previous studies have suggested that large strains may result from the bending deformation of the material itself (Mater. Horiz., 11, 1079, 2024; J. Appl. Phys. 134, 054101, 2023; Advanced Materials, 36, 2404682, 2024). This bending deformation is largely increased with reduced thickness. While the intrinsic deformation should not show a strong dependence on thickness. This is why I asked the author to measure the strain with thickness. According to the results, the strain is likely induced by bending. The authors need to carefully ensure the origins of strain since all the record-high data will be a metric to evaluate others' future works.

The authors should provide additional evidence to demonstrate that the observed strain is not due to bending deformation. For instance, more detailed images of the measurement setup and direct visual test would be useful. This is also the unaddressed question about the visual test for strain in my previous comment. In general, micro-scale deformation can be directly measured, if they are unable to do it, then the author at least should provide other experimental evidences to support it.

2) In the response to my previous comment (5) "A large negative strain value suggests a significant amount of irreversible non-180° domain switching with additional strain attributed from field induced phase transition". This is too general, the author needs to provide quantitatively analysis on the strain level, or if it can be interpreted by current understandings, the author needs to provide insight discussion on it. I don't think the phase transition based on the change of lattice parameters and non-180° domain switching can induce such a large strain. This is also why I asked the author to provide simulation results, which was also not addressed.

Reviewer #3

(Remarks to the Author)

It is recommended to accept.

Reviewer #4

(Remarks to the Author)

Version 2:

Reviewer comments:

Reviewer #2

(Remarks to the Author)

I agree with the authors that the strain shape of their sample should not originate from bending, especially in Fig. 2e. Hence, If the measurement is correct, this work is worth publication in Nature Communications. On the other hand, the authors failed to interpret the origin of such a high strain, I still have several comments listed before its publication.

1) It seems the authors are unable to provide any visualized measurement of the displacement of their samples. Then, the authors are suggested to include all the descriptions of why their strain is not due to bending in the supporting materials;

2) The authors generally attributed the high strain to the critical endpoint and morphotropic phase boundary, which claimed too much. They selected the MPB composition of PMN-PT as the base doped with Er. Then the MPB contributes to the higher strain is readable, however, the reviewer did not see any behaviors that can be related to the critical endpoint. The d_{33} does not show a critical value as an applied field (Fig. 5S) in the selected range. The dielectric constant and piezoelectric coefficients should reach the maximum at the CEP in the E-T-content phase diagram. If the authors like to claim the contribution from critical phenomena, then at least they should discuss how the CEP shifts with different Er content, otherwise, I do not suggest the authors claim too much about the mechanism.

In general, this work discovered an ultra-high electromechanical coupling in Er-doped PMN-PT, which is attractive in this field. However, I strongly suggest the author weaken their claims on the mechanism if they cannot interpret it clearly, both in the abstract and main text. In this way, they can leave some open questions for others to do more theoretical research.

Version 3:

Reviewer comments:

Reviewer #2

(Remarks to the Author)

The authors have adequately addressed my comments.

Responses to Reviewers' Comments

We thank the reviewers for their helpful and constructive comments and respond to their specific points below. Changes to the text are highlighted in the manuscript.

Reviewer #1 (Remarks to the Author):

Zhang et al reported the high-strain induced by small electric field in erbium-doped PMN-PT ceramics. The piezoelectric properties of this material are overall good, which exhibit high strain (3.19% bipolar and 0.8% unipolar) under a low applied field of 2 kV/mm, resulting in record-breaking piezoelectric coefficients (d_{33}^ values of 15950 and 4014 pm/V, respectively). Authors proposed a mechanism for such a superior property, namely, these exceptional properties stem from both the morphotropic phase boundary and differential distortion between the two phases present at the boundary.*

Although this work reports interesting and great materials performance that is the only selling point for a publication, I have concerns about the interpretation of the experimental results and the discussion of the mechanism, which I think should be further clarified. At current stage, the manuscript should be rejected and the manuscript should be rewritten.

1) It is not fair to overly emphasize the importance of low-driven field. Sometimes high-drive applications are also important.

Re: We recognize that high-drive applications are important in various advanced technological fields. Applications such as high-power actuators, transducers, and certain medical devices benefit significantly from high-driven field performance. These applications necessitate robust materials that can withstand and perform efficiently under such conditions.

In our study, we emphasized the superior performance of erbium doped PMN-PT ceramics under low-driven fields because of its relevance to specific applications where energy efficiency, miniaturization, and operational stability at lower voltages are critical. These include precision positioning systems, low power sensors, and portable electronic devices, where minimizing energy consumption is paramount. The high strain and exceptional piezoelectric coefficients achieved at low fields in our material make it particularly suitable for these applications.

To address the reviewer's concern, we have now revised the manuscript to provide a more balanced introduction that acknowledges the importance of high-driven field applications. The text on page 3 has been modified accordingly and an additional reference (no. 7) is included.

2) Lack of experimental evidence for proposed mechanism. Authors performed temperature dependence of structural analysis by neutron diffraction. Yet, I think an electric-field dependent structural analysis is more important here. Authors said there

is a differential distortion between the two phases present at the boundary. If they want to prove it, they should conduct in situ electric-field driven XRD or neutron diffraction experiments to observe the structural change under the small and large applied field.

Re: Based on the reviewer's suggestion, we have now conducted electric-field-dependent XRD measurements and present the results in Figure 11 of the supplementary file. The (111)_{pc} and (200)_{pc} diffraction peaks exhibit a noticeable shift toward higher 2θ angles as the electric field increases. When the electric field reaches 0.4 kV/mm, there is a rapid decrease in the T phase fraction and a corresponding increase in the O phase fraction, indicating an electric field induced phase transition from T to O phases. As the electric field continues to increase, the O phase fraction shows only a slight further increase. After removal of the electric field, the T phase exhibits a little recovery in its phase fraction with respect to the O phase. The unit cell parameters and unit cell volume show a clear decrease with electric field up to 0.4 kV/mm, which is close to the coercive field (ca. 0.45 kV/mm). A decrease in volume is consistent with thermodynamic considerations as the disordered higher symmetry T-phase transforms to the ordered lower symmetry O-phase. With further increases in the electric field, the cell parameters show little change. After removing the electric field, the poled sample did not fully recover to its original state, retaining reduced cell parameters. The reduced cell parameters after removal of the electric field are consistent with the P-E data, which show only a slight decrease from saturation polarization to remanent polarization upon the removal of the electric field. This phenomenon indicates a field-induced phase transition within the system, consistent with the observed strain variation, where large strain occurs when the electric field exceeds a certain value.

The experimental section on page 21 and the results and discussion on page 9 have been modified to include these new experimental results.

3) Furthermore, many ceramics and/single crystals are designed with MPB composition, most of which also have two or more phases. For various phases, the lattice distortion must be different. Authors cannot use such a normal concept to understand their discovery.

Re: The reviewer is correct in stating that many ceramics and single crystals are designed with MPB compositions and exhibit multiple phases with different lattice distortions. This is clearly not the only factor in determining whether the system will show high strain or not. The coexistence of the T and O phases at the MPB creates a highly responsive structure and it is the particular interplay between these phases under applied field in the present system that leads to significant lattice distortion and enhanced piezoelectric response. We have now expanded on the discussion of the mechanism of this phenomenon based on the new in situ XRD data which we now present as discussed in our response to the reviewer's second point above. The text on page 9 has been modified accordingly.

4) Authors should perform more detailed study on PNR and dielectric properties, also the loss behavior. Current PFM results are not enough. Similar PNR behavior are reported in many previous works.

Re: We acknowledge the necessity of a detailed investigation into the PNRs and dielectric behaviour and believe that we have presented much evidence to support the presence of PNRs. In addition to the PFM images, we have presented the direct observation of PNRs in the high resolution TEM images (Fig. 4f-i) and provide discussion of these based on PNRs. Additionally, the temperature dependence of dielectric permittivity and loss is presented in Supplementary Fig.3 showing the frequency dependence of dielectric permittivity and loss, consistent with the presence of PNRs. We also present the dielectric permittivity and loss behaviour of poled and unpoled samples in Supplementary Fig.10, which show the effect of poling on dielectric properties, including an anomaly in the dielectric spectrum associated with the field induced phase transition from T to O phases. Supplementary Fig.11 shows the Curie Weiss fitting of the dielectric spectrum and reveals a Burn's temperature, which is associated with the appearance of PNRs. As the reviewer indicates, PNRs have been observed in previous studies and we are not claiming this to be a novel feature. However, their presence does allow for rapid dipole switching and as such helps to facilitate the ease of the field induced transition, which as described in our response to point 3 above is one of the factors that contributes to the observation of high strain in this system.

5) The material system reported here is not new. Many works on similar or even same materials and their piezoelectric properties have been reported. It is interesting that why previous works have not observe the superior results reported here. I ask this question because authors said their samples were prepared by "conventional solid-state methods."

Re: It is true that rare earth element doping in PMN-PT systems has been widely studied, and different dopants can have varying effects on the piezoelectric properties. For instance, Sm-doped PMN-PT is known to exhibit a higher piezoelectric coefficient (d_{33}) compared to other rare earth-doped variants, even when prepared using conventional solid-state methods. [Nat. Mater. 17, 349–354 (2018).]

However, erbium-doped PMN-PT ceramics have been less frequently reported, particularly in the context of their electric field induced strain properties. Most studies on erbium doping in PMN-PT have focused on its single crystals or ceramics for transparent or fluorescent properties, often employing either single crystal growth techniques or modified solid-state methods, such as processing in an oxygen-rich atmosphere.

Whilst there have been measurements of piezoelectric and optical properties of Er-doped PMN-PT [Journal of Alloys and Compounds 927 (2022) 166969], to the best of our knowledge only two studies have presented strain data. In the work by Long et al. on single crystals of 2% Er-doped PMN-0.34PT, a high d_{33} was observed but with only moderate electric field induced strain. The work of Liang et al. [APPLIED PHYSICS LETTERS 109, 132904 (2016)], was mainly concerned with the optical properties and the significance of the high electric field induced unipolar strain observed in 2% Er-doped PMN-0.25PT transparent ceramics, beyond its optical properties was not discussed. In that work, the materials were also synthesised by conventional solid-state methods, but with subsequent sintering in oxygen followed by hot pressing. We now refer to the significance of this work in relation to the mechanism proposed in the present study. The text on page 4 has been modified accordingly.

Reviewer #2 (Remarks to the Author):

The authors designed a rare earth element doped PMN-PT near MPB ceramic for high-strain applications. Although the presented data are attractive in this field, the reliability of the performances and scientific understandings of the origins of the giant electro-strain need to be reinforced before publication in Nature Communications.

1) The first concern focuses on the strain data. Fig. 1e presents severely unrecovered strain in the field on/off cycles, which is unexpected in converse piezoelectric effect, for example, the ref. 9 they cited, showing a normal shape. Is there any explanation for this phenomenon? If the unrecovered strain cannot be used in a practicable application, then the claim they made, i.e. 0.8% strain is not accurate.

Re: Thank you for this insightful comment. In fact, the main contribution to strain in this system is the field induced transition and domain switching, with the converse piezoelectric effect, which shows linear behaviour, only making a small contribution. The sample indeed exhibits residual strain immediately after removal of the field, which is due to the kinetics of domain reorientation. This is also seen in Ref. 9, where poling in the $\langle 111 \rangle$ and $\langle 001 \rangle$ directions for PZN-8%PT single crystal results in hysteresis and hysteresis free strain behaviour, respectively. In that work the residual strain slowly returned to zero, a few seconds after removal of the electric field due to the depoling effect.

Similar behaviour is observed in the ceramic samples in the present study, with residual strain decreasing with time (see figure below). We agree that for practical applications, the recoverable strain is a critical parameter, but for classical ferroelectrics there will always be a residual strain, which requires time to recover. We refer to the value of 0.8% as being the maximum strain which is accurate and is the

same way that such strain is reported elsewhere [Science 378, 1125–1130 (2022), Adv. Mater., 30, 1705171, (2018)].

Unipolar strain behaviour of Er-PMNPT. After 10 seconds the field was removed and the strain slowly decays (indicated by red dashed circle).

However, in response to the reviewer's point 3 below, we have now carried out measurements on samples of different thickness and demonstrate that both the residual strain and the maximum strain vary with thickness. This could be an approach to tailoring the strain behaviour to achieve optimal performance. We now discuss this on page 7.

2) The authors should provide multiple cycle measurements of their samples, both unipolar and bipolar strain, seeing the stability of the performances in cyclic tests.

Re: We have now performed multi-cycle strain measurements for both unipolar and bipolar conditions, and the results are presented in the supplementary information (Fig. 8). The relevant discussion has been added to the manuscript on page 7. The strain shows very little change after 1000 cycles, indicating good stability in strain performance.

3) Both the bipolar and unipolar strain data are very impressive, I am wondering whether the author can provide the visual test, for example, a high-speed camera with sufficient resolution since its strain should be sufficiently high for capturing. Is there any thickness-dependent phenomenon, for example, electro-strain increases with reduced thickness?

Re: Based on the current strain data, it would be difficult to capture this effect visually even with a high-speed camera due to resolution limitations. For example, a 0.48 mm thick sample shows a maximum strain variation of approximately 3.5%, equivalent to 16.8 μm .

We have now conducted thickness-dependent strain measurements for the $x = 0.30$ composition and present the data in the supplementary information (Fig. 7). The strain clearly increases as the thickness decreases. This phenomenon can be explained by considering the reduced mechanical constraints at the surface to accommodate field induced distortion. Since the observed strain of the sample is a combination of bulk and surface strains, samples with a higher surface to bulk ratio exhibit higher values of total strain. These results are now discussed on page 7.

4) The ultra-high strain data needs more quantitative analysis. According to the authors' description, they attributed the strain to be induced by the phase transition between O and T phases. Phase transition could involve high strain, while they are lack of direct evidence that the transition occurs under electric fields. In-situ XRD measurement under various electric fields can provide proof.

Re: In-situ XRD analysis has now been conducted and are discussed in detail in our response to Reviewer 1's second comment. The results are presented and discussed on page 9.

5) On the other hand, the unipolar strain exhibits an ultra-high contraction strain during the domain switch, which is even higher than 3%. I do not think the phase transition can solely be interpreted. The difference in unit cell volume between the O and T phases is lower than 1% (fig, 3d), and the ceramics are untextured. What are the concrete differences in lattice parameters from T to O? The in-plane strains may also help understand the giant performances, I recommend the author to measure them.

Re: We agree that the phase transition alone cannot fully explain the observed ultra-high strain. We now provide much more detail on this mechanism. For a typical ferroelectric with butterfly S-E loops, the bipolar strain consists of both negative and positive components. Negative strain is defined as the difference between the minimum strain and the strain at zero electric field during bipolar cycles. The remanent strain at zero field represents the contribution of irreversible non-180° domain switching. A large negative strain value suggests a significant amount of irreversible non-180° domain switching with additional strain attributed from field induced phase transition. On the other hand, positive strain occurs when an electric field is applied to the poled sample, leading to additional strain. This process involves both domain switching and the piezoelectric effect. We now explain this in detail in the manuscript. The text on Page 5-6 has been modified accordingly and an alternative figure 1d is included to explain this feature.

We have now added the lattice parameters of O, T and C phases in the SI file.

We agree that in-plane strains measurement could help to evaluate the total volume changes during the field applied. Unfortunately, we currently lack facilities to perform such experiments.

6) Fig. 3b, 3c, 3d, and 3e present differences between O and T phases as a function of temperature, indicating the phase transition behaviors. Meanwhile, the author also claims their performance is attributed to the morphotropic phase boundary, which should be nearly independent of temperatures. The MPB can contain multiple energy degenerated phases, but it is very difficult to claim their contents and temperature-dependent concentrations. The authors should not simply attribute the performance to MPB. Such giant strain associated with phase transition is more like a critical phenomenon in ferroelectrics rather than MPB (Nature, 2006, 441(7096): 956-959).

Re: What we were claiming was that the compositional proximity to the MPB at ambient temperature is a contributing factor to the appearance of high strain. However, the reviewer is correct in that the situation is more nuanced in that it is not the MPB itself but the accessibility of phases of similar free energy, i.e. the possibility of a transition between structures showing differing degrees of distortion. We have altered the text on pages 11 to make this distinction clearer. As discussed above, we now include field dependent XRD measurements, which confirm the ease of the phase transition at relatively low fields, rather than relying solely on the thermal variation of phase behaviour to illustrate this.

We also acknowledge that critical phenomena in ferroelectric materials, such as the critical end point (CEP), could significantly contribute to the observed giant strain. The CEP is sensitive to compositional variation. The electric field induced ultrahigh strain in the $x = 0.30$ composition suggests that the CEP is close to room temperature. This proximity facilitates polarization rotation and the field induced phase transition, enabling easier alignment of polar nano regions and lowering the energy barrier for domain switching. This makes the materials highly responsive to the external electric field. The in-situ XRD results support this hypothesis, demonstrating that as the electric field reaches a critical field, the material undergoes substantial structural changes. We have now altered our discussion on pages 10-11 to reflect this and include the additional reference suggested by the reviewer.

7) The authors are expected to provide theoretical calculations, such as phase field simulation or density functional theory, to support the claim that the ultra-high strain comes from lattice strain due to phase transitions.

Re: We agree with the reviewer that theoretical calculations could be useful in establishing whether high strain is associated with the field induced phase transition.

However, based on our new results including in situ XRD and thickness dependent strain measurements, we now show that other factors such as surface effects, domain reorientation, internal mechanical clamping, and polar nano regions, play a crucial role in the overall strain behaviour. While theoretical calculations such as DFT are often used to explore the origins of ferroelectric and piezoelectric properties, these approaches cannot fully account for the complex external factors present in the current system. The accurate modelling of all these factors would represent a significant challenge. Given the intricate nature of these influences and the fact that our experimental data already strongly support our conclusions, we believe that such theoretical calculations are unlikely provide additional clarity or adequately reflect the multifaceted nature of the observed strain.

Reviewer #3 (Remarks to the Author):

The authors achieve attractive piezoelectric properties in the erbium-doped lead magnesium niobium titanate ceramics. However, some important issues have yet been explained.

1. Compared with the extremely high d_{33}^ and unipolar strain (under 2 kV/mm), the d_{33} value is quite low. The authors should explain it clearly.*

Re: The difference between the low d_{33} value and the extremely high d_{33}^* values can be explained by the nature of these measurements. The d_{33} value represents the piezoelectric response under a small, static electric field in the linear region of piezoelectric response, while d_{33}^* represents the effective piezoelectric response under larger electric fields, where non-linear effects become significant, including field-induced phase transitions and domain reorientation. Under small electric fields, the field induced phase transition between T and O phases is not active, resulting in a lower d_{33} value. However, at higher electric fields, this phase transition is activated, contributing substantially to the strain and d_{33}^* values. Additionally, higher electric fields can mobilize domain wall motion and domain reorientation, further enhancing d_{33}^* . We now make this point clearer on page 3.

2. How about the leakage behavior of the samples?

Re: Based on the I-E and P-E loops, we can confirm that the leakage current is small and can be considered negligible. Additionally, the loss tangent of the studied compositions, as presented in Figures 3-4 in the supplementary file, shows relatively low values at room temperature, further indicating minimal leakage and stable dielectric properties.

3. *The authors should compare the dielectric properties, including both the dielectric constant and dielectric loss for the samples under various frequency and temperature.*

Re: We have provided the temperature dependence of dielectric properties, including both the dielectric constant and dielectric loss, for samples under different frequencies. These data are presented in Supplementary Fig. 3 and are discussed on page 5.

4. *The authors report that the d_{33}^* decreases with increasing the field strength. Why does this happen?*

Re: As shown in supplementary Fig. 6, the unipolar strain increases with electric field strength, but the rate of increase slows down as the field strength continues to increase, eventually reaching a saturation point. This behaviour is primarily due to the reorientation of domains. As the electric field increases, the domains gradually align, and at a certain field strength, they approach a fully aligned state. Once this alignment nears completion, the strain increase rate diminishes, leading to a decrease in d_{33}^* with further increases in electric field. This type of behaviour is also observed in other relaxor materials, see for example [J. Appl. Phys. 82, 1804–1811 (1997)]. We have modified the text page 7 to explain this.

5. *The authors report that for the sample 0.30, the O phase could easily transform into T phase. Besides the evidence in distortion of the B-O2-Bo angle, is there any other evidences? Maybe, some simulation results should be added. Another question is whether it is easy to fabricate the sample 0.30 with so many metastable phases.*

Re: As shown in fig 3b, the phase fractions of O and T phases remain relatively stable between -173 and 7 °C. However, between 7 and 80 °C, phase transformation between O and T phases occurs as the T phase enters a region where it is the thermodynamically stable phase, before it itself transforms to the cubic phase at 140 °C. The thermal variation of the B-O2-B angle is a simple way of monitoring the structural distortion over the three polymorphs. We now additionally show the ease of phase transformation at applied fields with our new in situ XRD data, consistent with the results of dielectric measurements on poled and unpoled samples (Supplementary Fig. 13). The O → T transition in the PMNPT system is well known as is the coexistence of these phases at the MPB [Nat. Mater. 17, 349–354 (2018)].

The reviewer is right to question whether the synthesis of the $x = 0.30$ sample, with its multiple phases present, is easy. As outlined in the experimental section this is a relatively straightforward solid-state synthesis. We conducted parallel syntheses to ensure the repeatability and consistency of the fabrication process. Consistency was

observed not only in phase composition but also in properties across these batches. We now specifically state this on page 20.

6. For the PFM analyses, the contrast of the “-10 V/10 V” area is not 180/-180. Why does this happen? Is 10 V enough for poling the samples?

Re: The phase contrast for the ± 10 V regions is expected to be between 0° and 180° or 0° and -180° , rather than between 180° and -180° , as this would imply a total phase difference of 360° . As shown in Supplementary Figure 12, the phase versus voltage data indicates a total phase angle contrast of approximately 180° , with a coercive field of around 2 V. Therefore, applying ± 10 V is more than sufficient to fully switch the sample.

7. In the experimental section, the poling information should be provided.

Re: We thank the reviewer for spotting this omission. The poling process occurs during the PE measurement, where the samples are under an AC poling field of 5 kV/mm. We have changed the text on page 20 to make this clear.

Reviewer #4 (Remarks to the Author):

I co-reviewed this manuscript will co-reviewed this manuscript with one of the reviewers who provided the listed reports.

Re: There are no separate comments to respond to from this reviewer.

Responses to Reviewers' Comments

We thank the reviewers for their helpful and constructive comments and respond to their specific points below. Changes to the text are highlighted in the manuscript.

Reviewer #1 (Remarks to the Author):

The authors have addressed my concerns. It can be published in Nature Communications.

Re: We thank reviewer for their time and effort in reviewing the manuscript.

Reviewer #2 (Remarks to the Author):

The authors have partially addressed my concerns. However, I remain skeptical about the manuscript due to the lack of convincing answers to my major comments.

1) For ceramics, a strain exceeding 3% is likely to cause material fracture and failure unless it originates from a phase transformation (not an elastic process). However, the existing XRD evidence does not support such a significant strain. On the other hand, previous studies have suggested that large strains may result from the bending deformation of the material itself (Mater. Horiz., 11, 1079, 2024; J. Appl. Phys. 134, 054101, 2023; Advanced Materials, 36, 2404682, 2024). This bending deformation is largely increased with reduced thickness. While the intrinsic deformation should not show a strong dependence on thickness. This is why I asked the author to measure the strain with thickness. According to the results, the strain is likely induced by bending. The authors need to carefully ensure the origins of strain since all the record-high data will be a metric to evaluate others' future works.

The authors should provide additional evidence to demonstrate that the observed strain is not due to bending deformation. For instance, more detailed images of the measurement setup and direct visual test would be useful. This is also the unaddressed question about the visual test for strain in my previous comment. In general, micro-scale deformation can be directly measured, if they are unable to do it, then the author at least should provide other experimental evidences to support it.

Re: As we have shown in Fig. 3, there is a field induced transition supported by XRD data under applied field. However, the maximum field achievable in the XRD set is limited by the size of gap between the two electrodes which is governed by minimum size of the X-ray illumination window of 5 mm. Our set up the in situ XRD experiment is now included in the SI as Fig. S24a and for convenience is repeated below. As the

minim gap was 5 mm, we could not safely apply a high enough field to match the highest strain we tested in silicon oil for the S-E test. However, even under the low level of applied field/voltage, a field induced transition is observed.

Figure 1. Schematic diagram of XRD measurement under DC electric field.

We fully understand the concern of the referee that many recent popular papers show ultra-high strain, but their high strain is due to sample bending effects. As shown in the publications suggested by the referee, the high strain caused by bending effects in these works is associated with asymmetric S-E loops. Here, we show that our S-E loops for the samples of different thickness are symmetric, which suggests that high strain in our samples is not due to bending deformation. Our experimental setup for measuring S-E loops is shown in Fig. 2 below (now Fig. 24b in the SI). The ceramic sample was placed between a spherical top electrode and a flat bottom electrode. In the case of a bending deformation such a setup would be expected to produce asymmetric S-E loops according to He et al. (see fig 2b in *Mater. Horiz.*, 11, 1079, 2024, one of the references suggested by the referee).

Figure 2. Schematic diagram of test model for S-E loop measurement

We have modified the text on page 8 in order to emphasize this point.

We argue in the paper that the increase in measured strain with decreasing thickness, is attributable to the reduced mechanical constraints at the surface. To further validate this claim, we have now performed micro-indentation tests on samples of varying thickness, as shown in the figure below. The results indicate that thinner samples exhibit lower hardness corresponding to lower values of Young's modulus [B.R. Lawn, A.G. Evans, D.B. Marshall, J. Am. Ceram. Soc. 63 (1980) 574–581]. This reduced modulus lowers the mechanical clamping effect, making thinner samples more susceptible to deformation under an applied field. Consequently, the higher surface-area-to-volume ratio in thinner samples enhances their interaction with the applied electric field, facilitating higher strain. We have modified the text on pages 8, 12 and 13 and include Fig. 3 below in the SI as Fig. S10. The experimental text on page 20 has also been modified.

Figure 3. Thickness dependence of Vickers hardness for the $x = 0.30$ composition in the system $\text{Er}_{0.025}\text{Pb}_{0.09625}(\text{Mg}_{0.33}\text{Nb}_{0.67})_{1-x}\text{Ti}_x\text{O}_3$. Scale bar is 10 μm .

2) In the response to my previous comment (5) "A large negative strain value suggests a significant amount of irreversible non-180° domain switching with

additional strain attributed from field induced phase transition". This is too general, the author needs to provide quantitatively analysis on the strain level, or if it can be interpreted by current understandings, the author needs to provide insight discussion on it. I don't think the phase transition based on the change of lattice parameters and non-180° domain switching can induce such a large strain. This is also why I asked the author to provide simulation results, which was also not addressed.

Re: As we explained in our responses to the reviewer's previous comments, while theoretical calculations such as DFT are often used to explore the origins of ferroelectric and piezoelectric properties, these approaches cannot fully account for the complex external factors present in the current system. We have discussed the possibility of such calculations with our collaborators who agree that accurate modelling of these factors would represent a significant challenge.

The reviewer is correct that phase transition and non-180° domain switching alone cannot fully explain the high strain observed in the *S-E* loop and that sample thickness plays a major part. We have further analysed our data to more clearly show the thickness dependence of strain by looking specifically at displacement (see Fig. 4 below).

The negative strain decreases as the thickness increases. Although the strain induced by phase transition and domain switching can be intrinsically high, as discussed above thicker samples suppress deformation due to their higher Young's modulus, which results in stronger mechanical clamping. In contrast, thinner samples exhibit a higher surface-to-bulk ratio and lower Young's modulus, allowing for greater deformation under the same electric field.

To support this observation, we have added the plots below in the SI as Fig. 9 showing the actual displacement of the ceramic under both bipolar and unipolar fields. For samples with thicknesses greater than 0.58 mm, the displacement remains nearly constant for both positive and negative fields. However, for samples thinner than 0.58 mm, the displacement was significantly larger than the thicker sample and shows a gradual increase with decreasing thickness.

This suggests that the significantly high strain observed in thinner samples is strongly influenced by their lower Young's modulus, which reduces clamping effects and enhances the electric field-induced strain. We believe that these findings provide a more comprehensible understanding of the strain mechanism and demonstrate the critical role of sample thickness in the observed behaviour.

Figure 4. Displacement (Δl) under (a) bipolar and (f) unipolar electric field at room temperature for samples of the $x = 0.30$ composition of different thickness (mm) in the system $\text{Er}_{0.025}\text{Pb}_{0.09625}(\text{Mg}_{0.33}\text{Nb}_{0.67})_{1-x}\text{Ti}_x\text{O}_3$.

Reviewer #3 (Remarks to the Author):

It is recommended to accept.

Re: We thank reviewer for their time and effort in reviewing this manuscript

Reviewer #4 (Remarks to the Author):

I co-reviewed this manuscript with one of the reviewers who provided the listed reports. This is part of the Nature Communications initiative to facilitate training in peer review

Re: There are no separate comments to respond to from this reviewer.

Responses to Reviewer's Comments

We thank the reviewers for their helpful and constructive comments and respond to their specific points below. Changes to the text are highlighted in the manuscript.

Reviewer #2 (Remarks to the Author):

I agree with the authors that the strain shape of their sample should not originate from bending, especially in Fig. 2e. Hence, If the measurement is correct, this work is worth publication in Nature Communications. On the other hand, the authors failed to interpret the origin of such a high strain, I still have several comments listed before its publication.

1) It seems the authors are unable to provide any visualized measurement of the displacement of their samples. Then, the authors are suggested to include all the descriptions of why their strain is not due to bending in the supporting materials;

We have now added the relevant discussion in the caption to Fig. 24 in the supporting information to confirm the strain is not due to bending mechanism. We have also included additional references in the supporting material.

Supplementary Fig. 24 Schematic diagrams of setups used for (a) XRD measurements under DC electric field and (b) S-E loop measurements. Recent studies have reported ultra-high strain; however, in many cases, this strain has been attributed to bending effects⁷⁻¹⁰. High strain caused by bending is typically associated with asymmetric S-E loops. In contrast, our S-E loops for samples of varying thickness are symmetric, strongly indicating that the high strain observed in our samples is not due to bending deformation. The setup used for measuring the S-E loops is shown in (b). In this configuration, the ceramic sample was placed between a spherical top electrode and a flat bottom electrode. According to He *et al.*,¹⁰ such a setup would produce asymmetric S-E loops if bending deformation were a significant factor. Therefore, the symmetry of the S-E loops observed in our study confirms that the high strain is intrinsic to the material rather than a result of bending effects.

2) The authors generally attributed the high strain to the critical endpoint and morphotropic phase boundary, which claimed too much. They selected the MPB

composition of PMN-PT as the base doped with Er. Then the MPB contributes to the higher strain is readable, however, the reviewer did not see any behaviors that can be related to the critical endpoint. The d_{33} does not show a critical value as an applied field (Fig. 5S) in the selected range. The dielectric constant and piezoelectric coefficients should reach the maximum at the CEP in the E-T-content phase diagram. If the authors like to claim the contribution from critical phenomena, then at least they should discuss how the CEP shifts with different Er content, otherwise, I do not suggest the authors claim too much about the mechanism.

We have now removed the relevant discussion relative to the CEP in the system on page 12 and have modified the abstract accordingly.

Changes to the abstract

Original

These exceptional properties stem from a combination of factors including the sensitivity of polar nanoregions, in this relaxor ferroelectric system, to the applied field, the proximity of the system to the critical end point, the change in polarisation direction between the tetragonal and orthorhombic polymorphs at the field induced phase transition and the thickness of the sample.

Revised

These exceptional properties stem from a combination of factors including the sensitivity of polar nanoregions to the applied field in this relaxor ferroelectric system, the thickness of the sample, and the energetic availability of polymorphs with different polar structures where a change in polarisation direction occurs at the field induced phase transition.

Changes on Page 12

Original

The occurrence of such a phase transition, at which the free energies of the two polymorphs involved are equal, is dependent on composition, temperature and electric field. Kutnjak et al.³⁴ have described the dependency of the electromechanical response in relaxor ferroelectrics on temperature, field and composition in terms of a critical end point (CEP) at which there is a change from first order to supercritical behaviour where properties such as heat capacity and permittivity become more diffuse. They argue that it is the proximity to the CEP that primarily determines the electromechanical response rather than the proximity to the MPB. In the case of the $x = 0.30$ and 0.32 compositions in the present study, the CEP likely occurs at near ambient temperatures and at fairly low applied fields. Thirdly, while the two polymorphs

have equal free energies at the phase transition, they have significantly different polarisation vectors.

Revised

The occurrence of such a phase transition, at which the free energies of the two polymorphs involved are equal, is dependent on composition, temperature and electric field. Thirdly, while the two polymorphs have equal free energies at the phase transition, they have significantly different polarisation vectors.

In general, this work discovered an ultra-high electromechanical coupling in Er-doped PMN-PT, which is attractive in this field. However, I strongly suggest the author weaken their claims on the mechanism if they cannot interpret it clearly, both in the abstract and main text. In this way, they can leave some open questions for others to do more theoretical research.

We thank reviewer for their time and effort in reviewing this manuscript and have added an additional sentence to the end of the outlook on page 12 pointing out the possibility of further theoretical studies.

Additional Sentence in the Outlook on Page 12

This remarkable phenomenon merits further theoretical studies to clarify the physical mechanism behind the observed high strain.